# The Validity of Ultra-Short-Term Heart Rate Variability during Cycling Exercise

**DOI:** 10.3390/s23063325

**Published:** 2023-03-22

**Authors:** Yukiya Tanoue, Shihoko Nakashima, Tomohiro Komatsu, Miki Kosugi, Saki Kawakami, Shotaro Kawakami, Ryoma Michishita, Yasuki Higaki, Yoshinari Uehara

**Affiliations:** 1Institute for Physical Activity, Fukuoka University, Fukuoka 814-0180, Japan; 2Ritsumeikan-Global Innovation Research Organization, Ritsumeikan University, Kusatsu 525-8577, Japan; 3Institute of Advanced Research for Sport and Health Science, Research Organization of Science and Technology, Ritsumeikan University, Kusatsu 525-8577, Japan; 4Faculty of Sports and Health Science, Fukuoka University, Fukuoka 814-0180, Japan; 5Center for Preventive, Anti-Aging, and Regenerative Medicine, Fukuoka University Hospital, Fukuoka 814-0180, Japan

**Keywords:** exercise intensity, time-domain, frequency-domain, non-linear analysis

## Abstract

Ultra-short-term heart rate variability (HRV) has been validated in the resting state, but its validity during exercise is unclear. This study aimed to examine the validity in ultra-short-term HRV during exercise considering the different exercise intensities. HRVs of twenty-nine healthy adults were measured during incremental cycle exercise tests. HRV parameters (Time-, frequency-domain and non-linear) corresponding to each of the 20% (low), 50% (moderate), and 80% (high) peak oxygen uptakes were compared between the different time segments of HRV analysis (180 s (sec) segment vs. 30, 60, 90, and 120-sec segments). Overall, the differences (bias) between ultra-short-term HRVs increased as the time segment became shorter. In moderate- and high-intensity exercises, the differences in ultra-short-term HRV were more significant than in low intensity exercise. Thus, we discovered that the validity of ultra-short-term HRV differed with the duration of the time segment and exercise intensities. However, the ultra-short-term HRV is feasible in the cycling exercise, and we determined some optimal time duration for HRV analysis for across exercise intensities during the incremental cycling exercise.

## 1. Introduction

Heart rate variability (HRV) is the fluctuation of the instantaneous heartbeat over time [1]. HRV is traditionally used for non-invasive assessments of cardiac autonomic nervous activity [2]. It is well-established that HRV is associated with incident heart failure [3], cardiac mortality, all-cause mortality [4], and depression [5]. In addition to such clinical associations, HRV is also expected to be applicable in a wide range of practical situations, such as daily health monitoring [6,7], sports conditioning [8,9], and exercise prescription [10,11].

There are several methods of analysis to quantify HRV. The time- and frequency- domain are traditional HRV analysis method [2,12,13]. The time-domain indices indicate the variability in the period between the successive heartbeats. The representative variables in the time-domain analysis are the standard deviation of all RR intervals (SDNN), and the root mean square of successive RR interval differences (RMSSD). The frequency-domain indices of HRV can distinguish the several periodic variations in the successive heartbeats as frequency components. In frequency-domain analysis, two main frequency components of HRV have been traditionally used. Generally, the frequency band of 0.04–0.15 Hz is defined as the low-frequency (LF) component. The LF has been reported to be related to baroreflex [14,15] but have not yet been fully agreed upon [16]. The 0.15–0.4 Hz frequency band is defined as the high-frequency (HF) component and is believed to reflect parasympathetic nervous activity. The HF components is also well known to reflect the periodicity of respiration [17]. In addition to time- and frequency-domain analysis of HRV, based on the idea that regulation of HRV is complex, non-linear analysis has been applied to HRV analysis, which indicates the unpredictability of a time series of heartbeats [18]. The representative applied approaches include Detrended fluctuation analysis (DFA) and sample entropy (SampEn). The DFA is derived from fractal analysis, which shows self-similarity, and the short-term exponent α1 and the long-term exponent α2 characterize it. SampEn can quantify the complexity of the time series data and has been demonstrated to be reliable for short data set. Recently, the multiscale entropy with SampEn and the analysis of time irreversibility have been proposed as non-linear analysis, and these have been demonstrated to be able to detect a loss of complexity of RR time series with congestive heart failure, aging and external stimulus [19,20].

The recording duration used to calculate HRV parameters is one of the crucial factors in accurately quantifying HRV. A recording duration of ≥5 min has been recommended to standardize HRV recordings across different studies [12]. On the other hand, recently, the validity and reproducibility of recording or analyzing HRV below 5 min (ultra-short-term) have been investigated in many studies [21,22,23,24,25,26,27,28,29,30,31,32,33]. Munoz et al. indicated that it is sufficient to accurately measure both SDNN and RMSSD in the supine position for 120 s (sec) [25]. On the other hand, Burma et al. concluded that 240 sec for the time-domain parameters and 300 sec for the frequency-domain parameters were sufficient to obtain accurate values for each parameter in a standing position [32]. However, almost all of these reports have been limited to assessing HRV in the resting condition; thus, it is unclear whether ultra-short-term HRV is valid in non-static conditions such as exercise. During exercise, the heartbeat and physiological parameters may greatly vary, even for a few seconds or minutes. Therefore, if the recommended duration of the HRV recording is applied under non-static conditions like exercise, there is concern that the HRV may be too different from true state. On the other hand, there are also concerns that too short a duration of HRV recordings may lead to an inaccurate evaluation of HRV parameters. Recently, one study examined the ultra-short-term HRV recording during incremental exercise [34]. This study investigated the HRV for a 6-min incremental exercise session, including low and high exercise intensities, and it was determined that the minimum duration required for ultra-short-term HRV analyses during exercise was 10–180 sec. However, further detailed verifications are needed in exercise conditions because the state of HRV also differs with exercise intensities. For example, the values for the time- and frequency-domain parameters of HRV decrease progressively from low to moderate exercise intensity [35,36]. Over moderate intensity, however, the extent of decrease is a little, but the low level is maintained. Based on these findings, when assessing ultra-short-term HRV during exercise, examinations must take exercise intensity into account.

Recently, there have been many attempts to detect metabolic changes using HRV [10,11,37]. In situations where drastic physiological changes can be observed, such as during exercise, it is important to appropriately detect true changes in HRV with as little time lag as possible. Therefore, this study aimed to examine the time required for accurate analyses of ultra-short-term HRV during exercise, considering exercise intensity. This study would contribute to providing the appropriate and practical technic of HRV analysis during exercise in exercise training and clinical site.

## 2. Materials and Methods

### 2.1. Research Methodology

The present study aims to propose a valid time duration for HRV analysis in exercise conditions. In the following three steps, we compared five ultra-short-term segments (30, 60, 90, 120, and 180-sec segments). First, in each of the five segments, we confirmed the changes in mean values of HRV with increasing exercise intensity while comparing physiological parameter changes. Secondly, at three levels of exercise intensities (low, moderate, and high-intensity), the 180-sec segment was used as the reference duration to confirm the correlation coefficient with other time segments. Finally, we determined the differences between 180-sec and other time segments at each exercise intensity. The novelty of the present study was to reveal the valid time duration for HRV analysis across all the exercise intensities. This finding would contribute to accurate HRV analysis in non-static conditions, such as exercise.

### 2.2. Participants

In this report, twenty-nine healthy adults were included, and their data (age, 28.4 ± 12.0 years; height, 170.3 ± 8.1 cm; weight, 62.5 ± 8.5 kg; ${\dot{V}O}_{2\mathrm{peak}}$, 42.6 ± 7.1 mL/kg/min) were analyzed. This report was created using the data from two studies [38]. The exclusion criteria in the two studies were as follows: severe or progressive medical disorders; exercise restrictions; ongoing alpha- or beta-blocker therapy; and current smoking. The protocols of the two approved studies were registered with the University Hospital Medical Information Network Clinical Trial Registry (UMIN000039639 and UMIN000048962).

### 2.3. Ethical Statements

All participants gave their written informed consent to participate in the study. The two studies included in this report were conducted per the principles of the Declaration of Helsinki and approved by the Ethics Committee of Fukuoka University (Nos. 17-10-05 and 20-09-01).

### 2.4. The Cycling Exercise Test Procedure

On the day before the exercise test, all participants were asked to avoid strenuous exercise, alcohol consumption, and caffeine consumption. In addition, the participants were required to fast for 2–3 h before the exercise test; however, they were allowed to drink water. Following the arrival of participants in our laboratory, blood samples were obtained from their cubital vein using a venous catheter, after which they rested for more than 30 min. This study adopted the ramp incremental exercise test using a cycle ergometer (AEROBIKE 75XLⅢ, KONAMI, Tokyo, Japan; Corival cpet, Lode BV, Groningen, The Netherlands). The participants pedaled at 10 watts for 4 min as a warm-up, followed by a gradual increase in the exercise workload (10-watt, 15-watt, or 20-watt increments every minute) until they were exhausted. Participants were instructed to maintain pedaling frequency at 60 revolutions per min. Exhaustion was determined as the point at which the pedaling rate fell below 50 revolutions per min. The temperature and humidity of the rooms in which the exercise test was performed were 23–24 °C and 30–60%, respectively.

### 2.5. Physiological Parameter Measurements

The respiratory gases were measured during the incremental exercise test using a mass spectrometer (ARCO-2000, Arco System, Chiba, Japan) via the mixing chamber method [39]. At the same time as respiratory gas measurement, the tidal volume and respiratory rate were measured using a flow sensor. The peak oxygen uptake (${\dot{V}O}_{2\mathrm{peak}}$) was determined from the mean VO_2_ of the last 30–60 sec prior to exhaustion.

Before the exercise test, the blood lactate level was measured in a seated, resting position (Lactate Pro 2, LT-1730, Arkray, Kyoto, Japan). Additionally, the measurement was performed at the end of every 30 sec period or every minute-long period during exercise until the blood lactate level reached 4 mmol/L. Finally, the blood lactate measurement was performed 3 min after the end of the exercise test.

Intravenous blood samples to assess catecholamine levels were continuously collected from the catheter every minute during the exercise test. After that, the collected samples were centrifuged at 3500 rpm for 15 min at 4 °C and stored at −80 °C until the analysis. Plasma catecholamine concentrations were analyzed by two commercial blood analysis companies (SRL Inc., Tokyo, Japan, and LSI Medience Corporation, Tokyo, Japan).

### 2.6. Heart Rate Variability Analysis

R–R intervals during the incremental exercise test were measured using a 3-lead electrocardiogram system (sampling rate, 1000 Hz; HRR-01, Ahm electronics, Tokyo, Japan) and the Reflex-Meijin version 4.12 software program (Crosswell, Yokohama, Japan). Only one participant’s measurements were performed using another 3-lead electrocardiogram system (sampling rate, 1000 Hz; AC-301A, Ahm Electronics, Tokyo, Japan) because of a system malfunction. The frequency analysis was conducted using the Memcalc/Win version 2 software program (GMS, Tokyo, Japan). Before the frequency analysis, the artifacts and abnormal beats were excluded using that software program according to the following criteria: (1) if the heart rate was below 30 bpm and above 220 bpm, (2) if outliers were detected by comparing adjacent R–R data, (3) if the validity was not confirmed among the successive three R–R data. After that, the power spectral densities were calculated using the maximum entropy method and the non-linear least squares method [40,41]. The LF band was set at 0.04–0.15 Hz. The HF band was adopted as 0.15–0.83 Hz per our previous study [38] because we included the respiratory rate in the high-intensity exercise period and excluded the influences of the pedaling cadence during exercise. The participants were not asked to control their respiratory rates during the exercise test. In addition to LF, HF, and LF/HF ratio, we calculated the Heart rate/LF ratio using the obtained heart rate and the LF component from the HRV analysis. This parameter has been reported to correlate with noradrenaline changes during exercise [38]. The mean RR and HR were adopted the calculated values using the Memcalc/Win version 2 software program. In addition to the frequency-domain, the time-domain (SDNN and RMSSD) and non-linear parameters (DAF α1, α2 and SampEn) were calculated using Kubios HRV software (version 4.0, Biosignal Analysis and Medical Imaging Group, Department of Physics, University of Kuopio, Kuopio, Finland) [42]. The noise detection level and beat correction for the R–R intervals data were set to “medium” and “automatic collection”, respectively. In the non-linear analysis, the beat ranges of DFA α1 and α2 analysis were set to 4–16 beats and 17–64 beats, respectively. The embedding dimension and tolerance in the SampEn analysis were set to 2 beats and 0.2 × SD, respectively.

### 2.7. Additional Data Handling

In this report, the length of time segment to analyze HRV was set at 30, 60, 90, 120, or 180 sec. Additionally, to examine the difference in exercise intensity, we selected exercise intensities corresponding to three levels of peak oxygen uptakes (20%, 50%, and 80% ${\dot{V}O}_{2\mathrm{peak}}$) corresponding to the low, moderate, and high exercise intensity, respectively. Consequently, we analyzed the R–R interval data corresponding to the exercise intensities attained during each of the 180, 120, 90, 60, and 30-sec time segments (see Figure 1). In the present study, the reason for adopting 180 sec as longest time segment of HRV analyses was not to overlap the adopted R–R intervals data between exercise intensities as well as possible.

### 2.8. Missing Data

There were missing data related to physiological parameters as follows: blood lactate at 50% ${\dot{V}O}_{2\mathrm{peak}}$ (*n* = 28) and at 80% ${\dot{V}O}_{2\mathrm{peak}}$ (*n* = 20); adrenaline and noradrenaline levels at 20% ${\dot{V}O}_{2\mathrm{peak}}$ (*n* = 24), at 50% ${\dot{V}O}_{2\mathrm{peak}}$ (*n* = 23) and 80% ${\dot{V}O}_{2\mathrm{peak}}$ (*n* = 20). There were also missing data in HRV. The frequency-domain indices of HRV corresponding to 20% ${\dot{V}O}_{2\mathrm{peak}}$ in one participant were missing because of the lack of R–R data to analyze frequency components (*n* = 28). In addition, electrocardiogram data obtained using the AC-301A system was incompatible with the Kubios HRV version 4.0 software. Thus, the data pertaining to the time-domain and non-linear HRV parameters in one participant were missing (*n* = 28). Moreover, the DAF α1 and α2 were unable to analyze in the 30-sec segment at 20% ${\dot{V}O}_{2\mathrm{peak}}$ (*n* = 0), at 50% ${\dot{V}O}_{2\mathrm{peak}}$ (*n* = 4), and 80% ${\dot{V}O}_{2\mathrm{peak}}$ (*n* = 26), and in the 60-sec segment at 20% ${\dot{V}O}_{2\mathrm{peak}}$ (*n* = 26). The SampEn was also unable to analyze in the 30-sec segment at 20% ${\dot{V}O}_{2\mathrm{peak}}$ (*n* = 0), at 50% ${\dot{V}O}_{2\mathrm{peak}}$ (*n* = 7), and 80% ${\dot{V}O}_{2\mathrm{peak}}$ (*n* = 27).

### 2.9. Statistical Analysis

Descriptive data are presented as the mean ± SD. The normality of the data distribution was confirmed using the Shapiro–Wilk test. Based on previous recommendations [43], HRV data were converted to common logarithmic (log) forms when the absolute values of HRV were distributed non-normally. Nevertheless, if it was distributed non-normally, a non-parametric test was used instead. The mean physiological and HRV parameters were compared among three exercise intensities using the one-way ANOVA or the Friedman test. If these were statistically significant, pairwise comparisons were conducted using the Bonferroni correction. The validity of ultra-short-term HRV was examined according to the previous recommendation of analyses [43]. First, the correlation between the 180-sec segment and other time segments was confirmed using the Pearson product-moment correlation coefficient (*r*) or Spearman’s rank correlation coefficient (*ρ*). In this study, the criterion for a high correlation coefficient was set at ≥0.8. Then, the Bland-Altman analysis was conducted to confirm bias (the mean difference) and 95% limits of agreement (LOA; the mean difference ± 1.96 SD between two different ultra-short-term HRVs). Finally, the bias between two different ultra-short-term HRVs was quantified using Cohen’s d statistics. For the variables with skewed data distributions, the effect size was calculated using a formula as follows:
*r* = Z-value/√number of samples


The Z-value was calculated using the Wilcoxon signed-rank test. In this study, the criterion for small effect size was set at ≤0.5 [44]. The number of data points was unified in the analysis if either one of a given pair was missing. The statistical significance level of the two-sided *p*-values was set at less than 0.05. Statistical analyses were performed using the SPSS software ver. 28 (IBM SPSS, Chicago, IL, USA). The bias and 95% LOA calculations were conducted using the GraphPad Prism version 9.2.0 software (GraphPad Software, San Diego, CA, USA).

## 3. Results

### 3.1. Changes in Physiological Parameters during Exercise

Table 1 shows the workload, ventilation, and blood parameters in the three stages (20%, 50% , and 80% ${\dot{V}O}_{2\mathrm{peak}}$) during the exercise test. VO_2_, tidal volume, respiratory rate, blood lactate, and noradrenaline levels increased from 20% to 50% ${\dot{V}O}_{2\mathrm{peak}}$, and further increased to 80% ${\dot{V}O}_{2\mathrm{peak}}$. The adrenaline level did not differ significantly between 20% and 50% ${\dot{V}O}_{2\mathrm{peak}}$ but increased significantly from 50% to 80% ${\dot{V}O}_{2\mathrm{peak}}$.

### 3.2. The Mean HRV Values Change during Exercise

Figure 2, Figure 3 and Figure 4 show mean HRV values (log) during exercise. Refer to Appendix A for the absolute values of HRV parameters, the mean RR and HR. Log SDNN, log RMSSD, log LF and log HF components decreased from 20% to 50% ${\dot{V}O}_{2\mathrm{peak}}$ and further decreased to 80% ${\dot{V}O}_{2\mathrm{peak}}$ in all ultra-short-term cases (*p* < 0.05). Log LF/HF in the 180-sec segment did not differ significantly among the three exercise intensities. However, log LF/HF in the 120-sec, 90-sec, and 60-sec segments decreased at 80% ${\dot{V}O}_{2\mathrm{peak}}$ compared to 20% and 50% ${\dot{V}O}_{2\mathrm{peak}}$ (*p* < 0.05). Log LF/HF in the 30-sec segment decreased at 80% ${\dot{V}O}_{2\mathrm{peak}}$ compared to 20% ${\dot{V}O}_{2\mathrm{peak}}$ (*p* < 0.05). Log Heart rate/LF increased at 50% ${\dot{V}O}_{2\mathrm{peak}}$ more than at 20% ${\dot{V}O}_{2\mathrm{peak}}$ and increased at 80% ${\dot{V}O}_{2\mathrm{peak}}$ in all ultra-short-term segments (*p* < 0.05). Among the non-linear HRV parameters, DFA α1 in the 180-sec, 120-sec, and 90-sec segments increased from 20% ${\dot{V}O}_{2\mathrm{peak}}$ to 50% ${\dot{V}O}_{2\mathrm{peak}}$ but decreased at 80% ${\dot{V}O}_{2\mathrm{peak}}$ compared to 20% and 50% ${\dot{V}O}_{2\mathrm{peak}}$ (*p* < 0.05). DFAα1 decreased in the 60-sec segment at 80% ${\dot{V}O}_{2\mathrm{peak}}$ compared to 20% and 50% ${\dot{V}O}_{2\mathrm{peak}}$ (*p* < 0.05). SampEn decreased at 50% ${\dot{V}O}_{2\mathrm{peak}}$ compared to 20% ${\dot{V}O}_{2\mathrm{peak}}$ but increased at 80% ${\dot{V}O}_{2\mathrm{peak}}$ compared to 50% ${\dot{V}O}_{2\mathrm{peak}}$ in the 180-sec, 120-sec, 90-sec, 60-sec segments (*p* < 0.05). In contrast to DFA α1, DFA α2 increased with increasing exercise intensity in all ultra-short-term cases (*p* < 0.05). Due to missing data, the 30-sec segment of DFA α1, α2 and SampEn were not statistically analyzed.

### 3.3. The Correlation Coefficient

For the time-domain HRV parameters (log SDNN and log RMSSD), the correlation between the 180-sec segment and other ultra-short-term segments were high at all exercise intensities (≥0.8; *p* < 0.05) except for log SDNN between the 180-sec and 30-sec segments at 80% ${\dot{V}O}_{2\mathrm{peak}}$.

For frequency components, the log LF component exhibited highly correlation between the 180-sec and 120-sec, 90-sec segments at all exercise intensities (≥0.8; *p* < 0.05). The log HF component was indicated to be highly correlated between the 180-sec and 120-sec, 90-sec, and 60-sec segments at all exercise intensities (≥0.8; *p* < 0.05). In terms of the log LF/HF ratio, the correlation between 180-sec and 120-sec segments was high at 20% and 80% ${\dot{V}O}_{2\mathrm{peak}}$ (≥0.8; *p* < 0.05) but not at 50% ${\dot{V}O}_{2\mathrm{peak}}$. The log LF/HF ratio correlation between the 180-sec and 90-sec segments was high at 20% ${\dot{V}O}_{2\mathrm{peak}}$ (≥0.8; *p* < 0.05); however, its value at 50% and 80% ${\dot{V}O}_{2\mathrm{peak}}$ was below the criterion for a high correlation coefficient. The log Heart rate/LF indicated a high correlation between the 180-sec and 120-sec, 90-sec, and 60-sec segments at all exercise intensities (≥0.8; *p* < 0.05).

In non-linear parameters, the DFAα1 demonstrated a high correlation between the 180-sec and, 120-sec, 90-sec segments at 20% and 80% ${\dot{V}O}_{2\mathrm{peak}}$ (≥0.8; *p* < 0.05), but it was below the criterion at 50% ${\dot{V}O}_{2\mathrm{peak}}$. The DFAα2 showed a high correlation between the 180-sec and 120-sec segment at 20% ${\dot{V}O}_{2\mathrm{peak}}$ only (≥0.8; *p* < 0.05). SampEn exhibited a high correlation between the 180-sec and 120-sec segment at 20% and 50% ${\dot{V}O}_{2\mathrm{peak}}$ (≥0.8; *p* < 0.05). The 30-sec segment’s DFAα1, DFAα2 and SampEn at 20% ${\dot{V}O}_{2\mathrm{peak}}$ were not statistically analyzed due to missing data (Table 2).

Based on these results, the acceptable minimum time segment of HRV analysis according to the correlation coefficient were summarized in Table 3.

### 3.4. The Bias and LOA from the Different Time Segments of the HRV Analysis

In addition, the bias between two different ultra-short-term HRV parameters was quantified by the effect size (Table 4). The Bland-Altman plots for the 180-sec segment and other ultra-short-term HRV segments (120 sec, 90 sec, 60 sec, and 30 sec) are indicated in Appendix A.

For the log SDNN and log RMSSD values in the time-domain analysis, the bias and LOA at 20% ${\dot{V}O}_{2\mathrm{peak}}$ increased in the shorter time segments compared to the 180-sec segment. Moreover, the bias in all ultra-short-term segments at 50% and 80% ${\dot{V}O}_{2\mathrm{peak}}$ was larger than at 20% ${\dot{V}O}_{2\mathrm{peak}}$. The log RMSSD in the 120-sec segments at 20% ${\dot{V}O}_{2\mathrm{peak}}$ was small bias (effect size; ≤0.5) in comparison with the 180-sec segment.

In the log LF, HF and Heart rate/LF also, the bias and LOA at 20% ${\dot{V}O}_{2\mathrm{peak}}$ increased with the shorter time segment compared to 180-sec segment. Moreover, the bias in all ultra-short-term segments at 50% and 80% ${\dot{V}O}_{2\mathrm{peak}}$ were larger than at 20% ${\dot{V}O}_{2\mathrm{peak}}$. The log LF in 30–120-sec segments, log HF, and Heart rate/LF in 90–120 sec segments at 20% ${\dot{V}O}_{2\mathrm{peak}}$ were small bias (effect size; ≤0.5) compared to the 180-sec segment. The log LF/HF ratio did not yield consistent results.

Regarding DFAα1, the bias in all ultra-short-term segments was small at 20% , and 50% VO_2peak,_ but the biases at 80% ${\dot{V}O}_{2\mathrm{peak}}$ were more significant than at 20% and 50% ${\dot{V}O}_{2\mathrm{peak}}$. DFAα2 was not observed to yield consistent results. The SampEn between ultra-short-term segments were small bias (effect size; ≤0.5) except for the 30-sec segment at 50% ${\dot{V}O}_{2\mathrm{peak}}$.

Based on these results, the acceptable minimum time segment of HRV analysis according to the effect size were summarized in Table 3.

## 4. Discussion

In this study, we examined the validity of ultra-short-term HRV during exercise with respect to exercise intensity. Consequently, although the ultra-short-term segments of mean HRV indicated the similar changes in HRV parameters with increasing exercise intensity, we found differences in the validity based on the duration of HRV analysis time segment and the level of exercise intensity.

The changes in mean HRV parameters during exercise were almost consistent with the previous studies. The time-domain (SDNN and RMSSD) and frequency-domain indices (LF and HF components) have been reported to decrease during exercise in many previous studies [38,45,46,47,48,49]. In the present study, both domains of parameters during exercise similarly decreased in all ultra-short-term segments. However, the mean LF/HF change decreased or did not change with increases in exercise intensity. LF/HF has been traditionally used as the index of the sympathetic nervous activity or sympatho-vagal balance [50], but it has been suggested to be difficult to detect the autonomic nervous activity changes during exercise [45,48,51,52]. We confirmed that LF/HF changed, contradictory with catecholamine concentration change that increased during exercise. Thus, the results of the present study do not support the assumption that LF/HF reflects sympathetic nervous activity. Recently, we reported the Heart rate/LF ratio had exhibited a high correlation with noradrenaline concentration during exercise [38]. In the present study, we also examined the Heart rate/LF ratio changes during exercise and found that it conformed to similar results in our previous study [38]. Therefore, Heart rate/LF may be the alternative index for LF/HF as sympathetic nervous activity. In the non-linear parameters, DFA α1 increased from light to moderate exercise intensity and decreased from moderate to high. These changes in DFAα1 during exercise have also been observed in the previous reports [53,54]. However, the mean DFAα1 of the 60-sec segment did not change at low to moderate exercise intensity. Thus, the too short segment could exhibit different changes of DFAα1 during exercise with the previous studies.

The mean HRV values indicated similar changes during exercise in all ultra-short-term segments. Still, the trends seem to be not necessarily matched between the 180-sec segment and other ultra-short-term segments. The previous study has demonstrated similar results in stress test (a head-up tilt) [55]. In particular, the frequency-domain and DFAα1 and α2 of non-linear tended to be inferior in correlation in shorter time segments. In a previous study also, it was indicated to be similarly tended that the correlation diminished in shorter segment cases in particular frequency-domain compared to the time-domain [34]. Moreover, in addition to the short-time segment, we found that the higher exercise intensity increased data variance. In the present study, at moderate and high exercise intensity, the correlation coefficient was lower for the 30-sec segment than other ultra-short-term segments. Moreover, LOA was also large in the same case. It was particularly conspicuous in shorter time segment cases of the frequency-domain HRV parameters. It may be considered that the difference in the ratio of the period of the physiological changes for the length of time segment in HRV analysis affected. For example, the respiratory rate at high-intensity was 33.8 ± 7.1 breaths/min during high-intensity exercise. This impact on the respiratory rate could be over 50% for the 60 sec or 30-sec segments. During exercise, the HRV is slight in high-intensity exercise [38]. Even if HRV is minor, due to the short time segment of analysis, slight deviations in the respiratory cycle or the baroreflex during exercise could significantly affect HRV parameters in the frequency-domain analysis. Previous research has found a negative association between the coefficient of variation for respiratory rate and HRV recording duration, and they have proposed that short-term HRV recording (≥5 min) are less influenced by physiological changes than ultra-short-term recordings [32]. Moreover, the differences in HRV between the duration of analysis time segment may also be influenced by the deflection point of HRV during exercise. The change in HRV was reported to have deflection points around the ventilatory threshold corresponding to moderate exercise intensities [10]. The variability of R-R intervals seems to remain relatively large until the deflection point compared to over the deflection point. Therefore, for the moderate to high exercise intensity, the existence of the deflection point within the time segments of HRV analysis may have affected the differences between the 180-sec segment and other ultra-short-term HRV segments.

The differences (bias) between ultra-short-term HRVs increased as the time segment became shorter. It was particularly conspicuous at low exercise intensity compared to moderate to high exercise intensity. It is simply considered that the difference in the amount of heartbeat data could affect HRV analysis. The HRV is higher in lower exercise intensity, and it non-linearly decreases with an increase in HR [56,57]. Therefore, the 180-sec segment, which included the long prior period from the real-time, was higher than other ultra-short-term HRV values. The Heart rate/LF ratio in the 180-sec segment was lower than the others. Therefore, even with ultra-short-term HRV, when the time segment for the HRV analysis is more extended, it may be far from the actual autonomic activity during exercise.

Recently, HRV assessment has become more efficiently conducted with the spread of wearable devices. The previous study for athletes has been suggested to measure the ultra-short-term HRV before training in multiple days for monitoring condition and training adaptation [58]. Accordingly, the realistic measurement time seems to be an important factor in spreading HRV assessment. The validity of ultra-short-term HRV has been previously examined in the resting condition. Although the criteria for validity and some measurement conditions (posture, resting periods before recording, and so on) were different among previous studies, the acceptable results for the length of recording or analysis were reported for the various segments, including the 60 sec [29], 120 sec [25,27], 240 sec [32] sec in time-domain, and the 90–180 sec [29], 240 sec [27], 300 sec [32] in the frequency-domain, 90–240 sec in non-linear analysis [29]. In exercise situations also, the ultra-short-term HRV is seemed to be important for accurately detecting real-time physiological change. In contrast to resting conditions, to our knowledge, one study has attempted examination under the non-static condition, such as exercise. This study concluded that the minimum time needed for the ultra-short-term HRV analysis was 10–180 sec in exercise conditions [34]. However, this previous study limit to did not consider the exercise intensity. Our data demonstrated that the validity of the ultra-short-term segment for HRV analysis was different for each exercise intensity. Therefore, we consider that it is vital to determine the time segment that can consistently obtain valid data across low to high exercise intensities.

There are several limitations to the present study. First, the present study adopted an incremental exercise test with the ramp method. Therefore, the validity of ultra-short-term HRV during steady-state exercise was unclear. The steady-state exercise might obtain stable HRV data that excluded the effect of non-linear changes observed in the ramp method. In addition, it was also unclear whether there was a difference in the 300 sec (5-min) segment. The time segment length in HRV measurement and analysis is recommended at 300 sec [12]. However, the present study did not compare the 300-sec segment because the segment time overlapped between each exercise intensity. The ultra-short-term HRV between different exercise intensities needs to be examined in future studies, including a 300-sec segment.

## 5. Conclusions

The validity of ultra-short-term HRV was different for exercise intensities. However, the ultra-short-term HRV is feasible in the cycling exercise, and we determined some optimal time duration for HRV analysis for across exercise intensities during the incremental cycling exercise in particular.

## Figures and Tables

**Figure 1 sensors-23-03325-f001:**
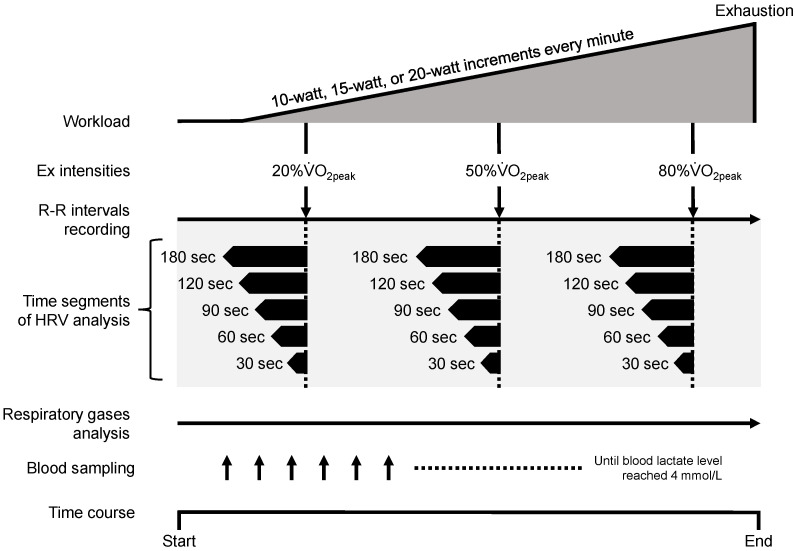
The experimental protocol and the adoption of HRVs corresponding to each level of exercise intensity. The black range in the time segment of HRV analysis indicates the time length of HRV analysis from each exercise intensity (20%, 50%, or 80% ${\dot{V}O}_{2\mathrm{peak}}$). All HRV analyses for an individual participant were performed using the same R–R intervals recording data. The up arrow indicates the sampling point of blood. Ex, exercise; HRV, heart rate variability; sec, seconds; ${\dot{V}O}_{2\mathrm{peak}}$, peak oxygen uptake.

**Figure 2 sensors-23-03325-f002:**
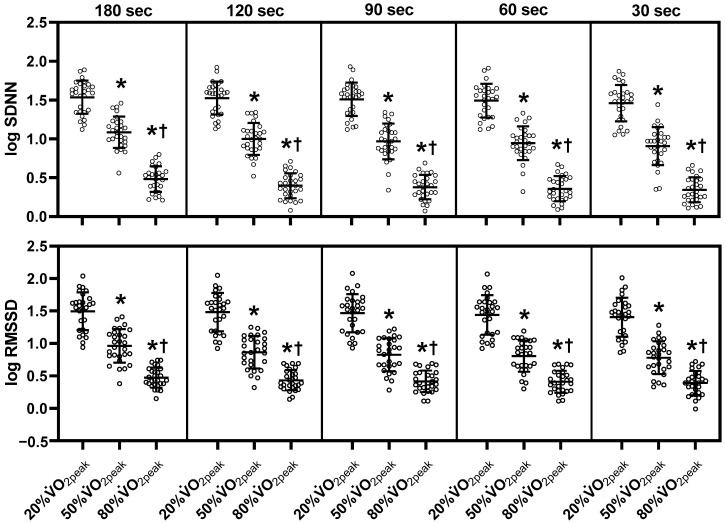
Changes of each ultra-short-term segment of time-domain HRV parameters during exercise. SDNN, standard deviation of all RR intervals; RMSSD, root mean square of successive RR interval differences; log, logarithmic; sec, seconds; ${\dot{V}O}_{2\mathrm{peak}}$, peak oxygen uptake. The middle line indicates the mean value. The top and bottom lines indicate the SD. The white plots indicate the individual data of participants. Note that there is missing data (see the Materials and Methods section). *, *p* < 0.05 vs. 20% ${\dot{V}O}_{2\mathrm{peak}}$; †, *p* < 0.05 vs. 50% ${\dot{V}O}_{2\mathrm{peak}}$.

**Figure 3 sensors-23-03325-f003:**
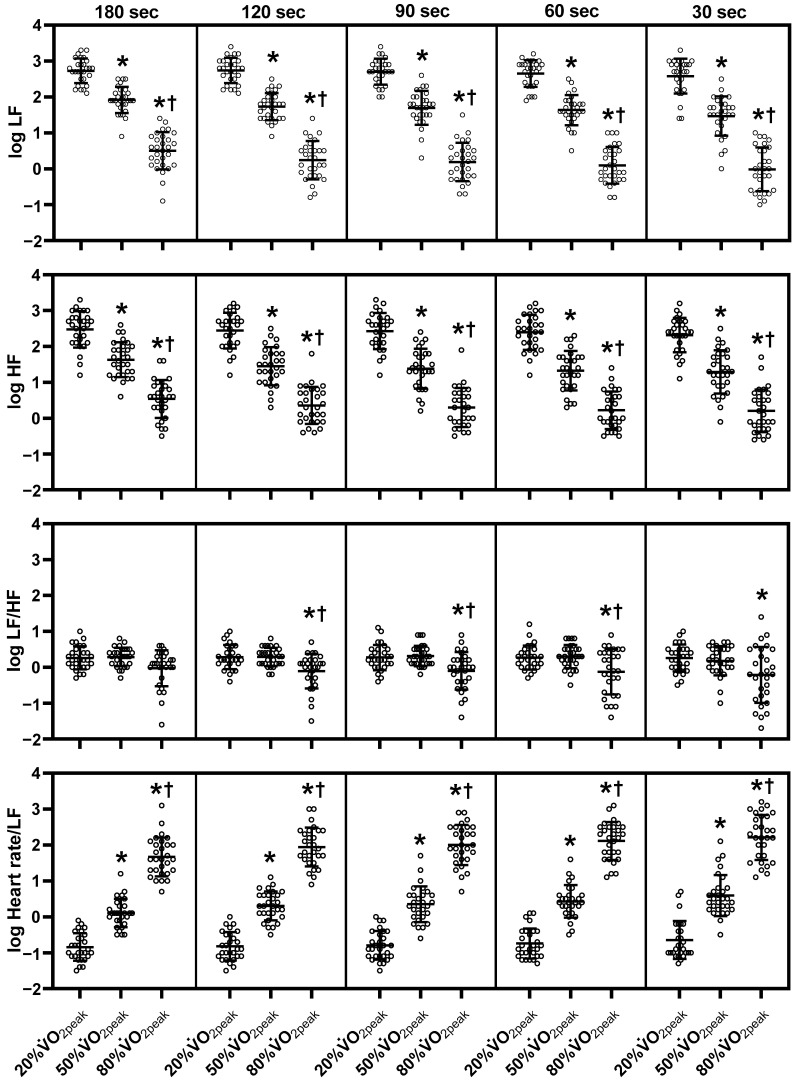
Changes of each ultra-short-term segment of frequency-domain HRV parameters during exercise. HF, high-frequency power; LF, low-frequency power; log, logarithmic; sec, seconds; ${\dot{V}O}_{2\mathrm{peak}}$, peak oxygen uptake. The middle line indicates the mean value. The top and bottom lines indicate the SD. The white plots indicate the individual data of participants. Note that there is missing data at 20% ${\dot{V}O}_{2\mathrm{peak}}$ (see the Materials and Methods section). *, *p* < 0.05 vs. 20% ${\dot{V}O}_{2\mathrm{peak}}$; †, *p* < 0.05 vs. 50% ${\dot{V}O}_{2\mathrm{peak}}$.

**Figure 4 sensors-23-03325-f004:**
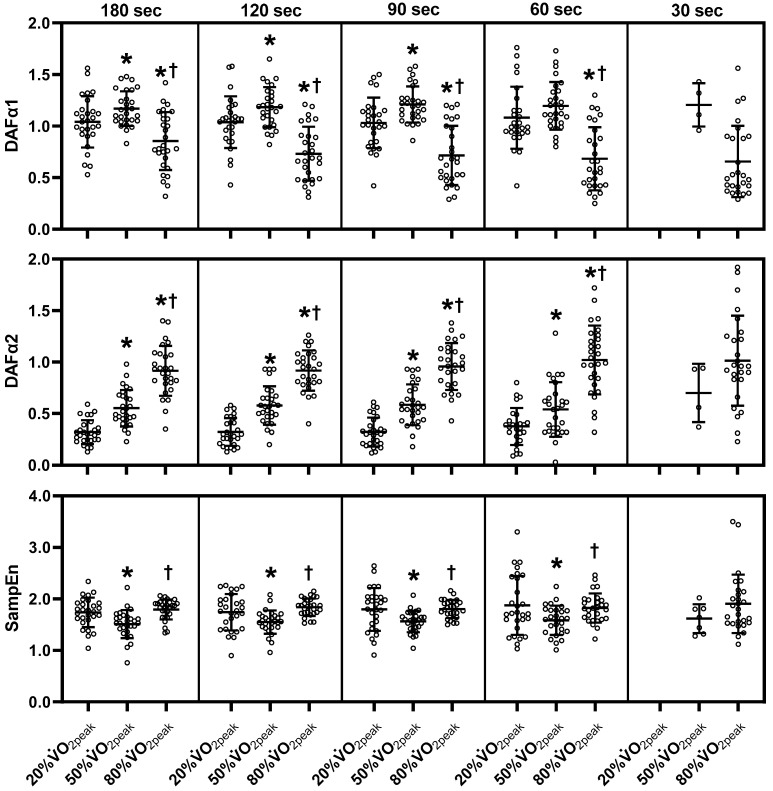
Changes of each ultra-short-term segment of non-linear HRV parameters during exercise. DFA, detrended fluctuation analysis; SampEn, sample entropy; log, logarithmic; sec, seconds; ${\dot{V}O}_{2\mathrm{peak}}$, peak oxygen uptake. The middle line indicates the mean value. The top and bottom lines indicate the SD. The white plots indicate the individual data of participants. Note that there is missing data (see the Materials and Methods section). *, *p* < 0.05 vs. 20% ${\dot{V}O}_{2\mathrm{peak}}$; †, *p* < 0.05 vs. 50% ${\dot{V}O}_{2\mathrm{peak}}$.

**Table 1 sensors-23-03325-t001:** The physiological parameters during exercise.

Variable	20% ${\dot{V}O}_{2\mathrm{peak}}$	50% ${\dot{V}O}_{2\mathrm{peak}}$	80% ${\dot{V}O}_{2\mathrm{peak}}$
Workload (watts)	29 ± 11	100 ± 21 *	173 ± 30 *^†^
VO_2_ (ml/kg/min)	8.5 ± 1.4	21.3 ± 3.5 *	34.1 ± 5.7 *^†^
Vt (L)	0.8 ± 0.2	1.3 ± 0.3 *	1.8 ± 0.4 *^†^
Respiratory rate (breath/min)	20.4 ± 3.7	24.4 ± 5.3 *	33.8 ± 7.1 *^†^
Blood lactate (mmol/L)	1.0 ± 0.3	1.5 ± 0.4 *	3.6 ± 0.7 *^†^
Adrenaline (ng/mL)	0.05 ± 0.04	0.07 ± 0.05	0.21 ± 0.25 *^†^
Noradrenaline (ng/mL)	0.42 ± 0.16	0.59 ± 0.23 *	1.15 ± 0.75 *^†^

Values are presented as the mean ± SD. Note that there were missing data pertaining to blood lactate, adrenaline, and noradrenaline (see the Materials and Methods section). VO_2_, oxygen uptake; ${\dot{V}O}_{2\mathrm{peak}}$, peak oxygen uptake; Vt, tidal volume. *, *p* < 0.05 vs. 20% ${\dot{V}O}_{2\mathrm{peak}}$; †, *p* < 0.05 vs. 50% ${\dot{V}O}_{2\mathrm{peak}}$.

**Table 2 sensors-23-03325-t002:** The coefficient of correlation between 180 sec and other ultra-short-term segments.

Variable	Exercise Intensity	Coefficient of Correlation			
		180 sec vs. 120 sec	180 sec vs. 90 sec	180 sec vs. 60 sec	180 sec vs. 30 sec
Time-domain					
log SDNN	20% ${\dot{V}O}_{2\mathrm{peak}}$	0.993 *	0.972 *	0.953 *	0.906 *
	50% ${\dot{V}O}_{2\mathrm{peak}}$	0.938 *	0.917 *	0.895 *	0.832 *
	80% ${\dot{V}O}_{2\mathrm{peak}}$	0.917 *	0.890 *	0.831 *	0.792 *
log RMSSD	20% ${\dot{V}O}_{2\mathrm{peak}}$	0.996 *	0.981 *	0.957 *	0.917 *
	50% ${\dot{V}O}_{2\mathrm{peak}}$	0.960 *	0.936 *	0.921 *	0.884 *
	80% ${\dot{V}O}_{2\mathrm{peak}}$	0.950 *	0.918 *	0.884 *	0.835 *
Frequency-domain					
log LF	20% ${\dot{V}O}_{2\mathrm{peak}}$	0.982 *	0.936 *	0.926 *^a^	0.814 *^a^
	50% ${\dot{V}O}_{2\mathrm{peak}}$	0.883 *	0.867 *	0.797 *	0.655 *
	80% ${\dot{V}O}_{2\mathrm{peak}}$	0.906 *	0.898 *	0.818 *	0.738 *
log HF	20% ${\dot{V}O}_{2\mathrm{peak}}$	0.992 *	0.969 *	0.950 *	0.910 *
	50% ${\dot{V}O}_{2\mathrm{peak}}$	0.934 *	0.908 *	0.819 *	0.730 *
	80% ${\dot{V}O}_{2\mathrm{peak}}$	0.911 *	0.859 *	0.836 *	0.825 *
log LF/HF	20% ${\dot{V}O}_{2\mathrm{peak}}$	0.959 *	0.868 *	0.735 *^a^	0.476 *^a^
	50% ${\dot{V}O}_{2\mathrm{peak}}$	0.799 *	0.736 *	0.725 *	0.362
	80% ${\dot{V}O}_{2\mathrm{peak}}$	0.821 *^a^	0.771 *^a^	0.769 *^a^	0.725 *^a^
log Heart rate/LF	20% ${\dot{V}O}_{2\mathrm{peak}}$	0.984 *	0.944 *	0.924 *	0.837 *
	50% ${\dot{V}O}_{2\mathrm{peak}}$	0.895 *	0.878 *	0.815 *	0.693 *
	80% ${\dot{V}O}_{2\mathrm{peak}}$	0.911 *	0.903 *	0.829 *	0.747 *
Non-linear analysis					
DFAα1	20% ${\dot{V}O}_{2\mathrm{peak}}$	0.942 *	0.839 *	0.717 *^a^	-
	50% ${\dot{V}O}_{2\mathrm{peak}}$	0.703 *	0.525 *	0.549 *	0.083
	80% ${\dot{V}O}_{2\mathrm{peak}}$	0.906 *	0.911 *^a^	0.864 *^a^	0.733 *^a^
DFAα2	20% ${\dot{V}O}_{2\mathrm{peak}}$	0.818 *	0.673 *	0.364	-
	50% ${\dot{V}O}_{2\mathrm{peak}}$	0.698 *	0.570 *	0.474 *	−0.946
	80% ${\dot{V}O}_{2\mathrm{peak}}$	0.529 *	0.218	0.403 *	−0.116
SampEn	20% ${\dot{V}O}_{2\mathrm{peak}}$	0.898 *	0.768 *	0.707 *	-
	50% ${\dot{V}O}_{2\mathrm{peak}}$	0.846 *	0.739 *	0.731 *	−0.036
	80% ${\dot{V}O}_{2\mathrm{peak}}$	0.605 *^a^	0.517 *^a^	0.319 ^a^	0.332 ^a^

Values are coefficient. The coefficient of correlation was calculated with Pearson (*r*). ^a^, the coefficient of correlation was calculated using Spearman (*ρ*). * *p* < 0.05. SDNN, standard deviation of all NN intervals; RMSSD, root mean square of successive RR interval differences, LF, low-frequency power; HF, high-frequency power; DFA, Detrended fluctuation analysis; SampEn, sample entropy.

**Table 3 sensors-23-03325-t003:** The acceptable minimum time segment of HRV analysis in exercise condition according to the statistical metrics.

Variable	Exercise Intensity	Statistical Metrics	
		Correlation Coefficient	Effect Size (Bias)
Time-domain			
log SDNN	20% ${\dot{V}O}_{2\mathrm{peak}}$	30 sec	n/a
	50% ${\dot{V}O}_{2\mathrm{peak}}$	30 sec	n/a
	80% ${\dot{V}O}_{2\mathrm{peak}}$	60 sec	n/a
log RMSSD	20% ${\dot{V}O}_{2\mathrm{peak}}$	30 sec	120 sec
	50% ${\dot{V}O}_{2\mathrm{peak}}$	30 sec	n/a
	80% ${\dot{V}O}_{2\mathrm{peak}}$	30 sec	n/a
Frequency-domain			
log LF	20% ${\dot{V}O}_{2\mathrm{peak}}$	30 sec	30 sec
	50% ${\dot{V}O}_{2\mathrm{peak}}$	90 sec	n/a
	80% ${\dot{V}O}_{2\mathrm{peak}}$	60 sec	n/a
log HF	20% ${\dot{V}O}_{2\mathrm{peak}}$	30 sec	90 sec
	50% ${\dot{V}O}_{2\mathrm{peak}}$	60 sec	n/a
	80% ${\dot{V}O}_{2\mathrm{peak}}$	30 sec	n/a
log LF/HF	20% ${\dot{V}O}_{2\mathrm{peak}}$	90 sec	30 sec
	50% ${\dot{V}O}_{2\mathrm{peak}}$	n/a	30 sec
	80% ${\dot{V}O}_{2\mathrm{peak}}$	120 sec	30 sec
log Heart rate/LF	20% ${\dot{V}O}_{2\mathrm{peak}}$	30 sec	90 sec
	50% ${\dot{V}O}_{2\mathrm{peak}}$	60 sec	n/a
	80% ${\dot{V}O}_{2\mathrm{peak}}$	60 sec	n/a
Non-linear analysis			
DFAα1	20% ${\dot{V}O}_{2\mathrm{peak}}$	90 sec	60 sec
	50% ${\dot{V}O}_{2\mathrm{peak}}$	n/a	30 sec
	80% ${\dot{V}O}_{2\mathrm{peak}}$	60 sec	n/a
DFAα2	20% ${\dot{V}O}_{2\mathrm{peak}}$	120 sec	60 sec
	50% ${\dot{V}O}_{2\mathrm{peak}}$	n/a	30 sec
	80% ${\dot{V}O}_{2\mathrm{peak}}$	n/a	30 sec
SampEn	20% ${\dot{V}O}_{2\mathrm{peak}}$	120 sec	60 sec
	50% ${\dot{V}O}_{2\mathrm{peak}}$	120 sec	60 sec
	80% ${\dot{V}O}_{2\mathrm{peak}}$	n/a	30 sec

Values are the acceptable minimum time segment of HRV analysis by each statistical metric. Decision criteria in each statistical metric are as follows: Correlation coefficient, ≥0.8; Effect size of mean difference (bias), ≤0.5; n/a, not applicable. SDNN, standard deviation of all NN intervals; RMSSD, root mean square of successive RR interval differences, LF, low-frequency power; HF, high-frequency power; DFA, Detrended fluctuation analysis; SampEn, sample entropy.

**Table 4 sensors-23-03325-t004:** The effect size of bias between 180 sec segment and other ultra-short-term.

Variable	Exercise Intensity	Effect Size			
		180 sec vs. 120 sec	180 sec vs. 90 sec	180 sec vs. 60 sec	180 sec vs. 30 sec
Time-domain					
log SDNN	20% ${\dot{V}O}_{2\mathrm{peak}}$	0.502	0.552	0.659	0.775
	50% ${\dot{V}O}_{2\mathrm{peak}}$	1.212	1.287	1.445	1.331
	80% ${\dot{V}O}_{2\mathrm{peak}}$	1.266	1.367	1.309	1.312
log RMSSD	20% ${\dot{V}O}_{2\mathrm{peak}}$	0.475	0.526	0.636	0.754
	50% ${\dot{V}O}_{2\mathrm{peak}}$	1.354	1.496	1.554	1.487
	80% ${\dot{V}O}_{2\mathrm{peak}}$	0.834	0.862	0.810	0.798
Frequency-domain					
log LF	20% ${\dot{V}O}_{2\mathrm{peak}}$	0.162	0.310	0.435 ^a^	0.491 ^a^
	50% ${\dot{V}O}_{2\mathrm{peak}}$	1.011	0.924	1.107	1.062
	80% ${\dot{V}O}_{2\mathrm{peak}}$	1.122	1.328	1.331	1.224
log HF	20% ${\dot{V}O}_{2\mathrm{peak}}$	0.417	0.389	0.523	0.741
	50% ${\dot{V}O}_{2\mathrm{peak}}$	1.027	1.140	0.993	0.835
	80% ${\dot{V}O}_{2\mathrm{peak}}$	0.812	0.859	1.044	0.986
log LF/HF	20% ${\dot{V}O}_{2\mathrm{peak}}$	−0.172	−0.049	−0.065 ^a^	−0.043 ^a^
	50% ${\dot{V}O}_{2\mathrm{peak}}$	−0.061	−0.192	−0.100	−0.250
	80% ${\dot{V}O}_{2\mathrm{peak}}$	0.343 ^a^	0.291 ^a^	0.251 ^a^	0.315 ^a^
log Heart rate/LF	20% ${\dot{V}O}_{2\mathrm{peak}}$	−0.218	−0.360	−0.594	−0.620
	50% ${\dot{V}O}_{2\mathrm{peak}}$	−1.077	−1.004	−1.198	−1.143
	80% ${\dot{V}O}_{2\mathrm{peak}}$	−1.174	−1.401	−1.409	−1.298
Non-linear analysis					
DFAα1	20% ${\dot{V}O}_{2\mathrm{peak}}$	0.036	0.072	0.127 ^a^	-
	50% ${\dot{V}O}_{2\mathrm{peak}}$	−0.085	−0.239	−0.125	0.477
	80% ${\dot{V}O}_{2\mathrm{peak}}$	1.040	−0.775 ^a^	−0.753 ^a^	−0.705 ^a^
DFAα2	20% ${\dot{V}O}_{2\mathrm{peak}}$	−0.034	−0.009	−0.267	-
	50% ${\dot{V}O}_{2\mathrm{peak}}$	−0.187	−0.190	0.052	0.112
	80% ${\dot{V}O}_{2\mathrm{peak}}$	−0.008	−0.142	−0.322	−0.143
SampEn	20% ${\dot{V}O}_{2\mathrm{peak}}$	−0.028	−0.224	−0.322	-
	50% ${\dot{V}O}_{2\mathrm{peak}}$	−0.276	−0.276	−0.356	−0.597
	80% ${\dot{V}O}_{2\mathrm{peak}}$	−0.181 ^a^	−0.034 ^a^	−0.052 ^a^	−0.097 ^a^

Values are the effect size. The effect size was calculated with Cohen’s d. ^a^, the effect size was calculated using Z-value from Wilcoxon signed-rank test. SDNN, standard deviation of all NN intervals; RMSSD, root mean square of successive RR interval differences, LF, low-frequency power; HF, high-frequency power; DFA, Detrended fluctuation analysis; SampEn, sample entropy.

## Data Availability

Data are not publicly available as they contain confidential clinical information of the subjects; however, certain parts of the data could be obtained from the corresponding author upon reasonable request.

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
