# Peer review of "The Validity of Ultra-Short-Term Heart Rate Variability during Cycling Exercise"

_sensors, 2023, doi:10.3390/s23063325_

Round 1
Reviewer 1 Report (New Reviewer)
I have read this manuscript with interest. It can be suitable for publication in sensors. However, there are some conceptual issues regarding HRV that authors must correct before publishing their manuscript. Once the corrections are made, the manuscript will present quality enough to be published.
L48-49. Authors state that LF reflects both sympathetic and Parasympathetic nervous activity. How only one variable could represent two opposite things? In addition, LF has been shown not to reflect any branch of nervous activity, thus this statement is not solid and must be deleted. It happens also in lines 51-52 where the authors state the relationship of HF and LF with baroreflexes. Authors should check this reference
Martelli, D., Silvani, A., McAllen, R. M., May, C. N., & Ramchandra, R. (2014). The low frequency power of heart rate variability is neither a measure of cardiac sympathetic tone nor of baroreflex sensitivity. American Journal of Physiology-Heart and Circulatory Physiology, 307(7), H1005-H1012.
L 57-58. Authors affirm that SD1 and SD2 are approaches for the analysis of non-linear dynamics. This statement is false as SD1 and SD2 are linear and within the time-domain. Non-linear analysis can be performed by the use of fractal analysis and entropy (Sample entropy mainly). The explanation of the Poincaré Plot in L59-61 is correct but it is not a non-linear analysis. Authors should check this reference
Ciccone, A. B., Siedlik, J. A., Wecht, J. M., Deckert, J. A., Nguyen, N. D., & Weir, J. P. (2017). Reminder: RMSSD and SD1 are identical heart rate variability metrics. Muscle & nerve, 56(4), 674-678.
Authors must mention the multiscale entropy and time irreversibility analysis as a kind of non-linear method of analysis. You can mention these studies
Abellan-Aynes, O., Naranjo-Orellana, J., Manonelles, P., & Alacid, F. (2020). MULTISCALE ENTROPY AND MULTISCALE TIME IRREVERSIBILITY ANALYSIS OF RR TIME SERIES DEPENDING ON AMBIENT TEMPERATURE. Journal of Mechanics in Medicine and Biology, 20(05), 2050029.
Costa, M., Goldberger, A. L., & Peng, C. K. (2005). Multiscale entropy analysis of biological signals. Physical review E, 71(2), 021906.
L85. It is true that exercise reduces HRV drastically. From moderate intensity, SDNN and RMSSD can remain close to zero. You should add some citations to state this, I recommend these studies
Abellán-Aynés, O., López-Plaza, D., Alacid, F., Naranjo-Orellana, J., & Manonelles, P. (2019). Recovery of heart rate variability after exercise under hot conditions: the effect of relative humidity. Wilderness & Environmental Medicine, 30(3), 260-267.
Michael, S., Graham, K. S., & Davis, G. M. (2017). Cardiac autonomic responses during exercise and post-exercise recovery using heart rate variability and systolic time intervals—a review. Frontiers in physiology, 8, 301.
Section "2. Research methodology" must be integrated into the material and methods section.
L183-184. SD1 and SD2 as non-linear. The changes regarding this issue must be correct throughout the whole manuscript.
L189. Why didn't you measure SampEn?
Table 2. As SD1 and RMSSD are exactly the same variable (RMMSD=SD1/Root square (2)) and SD2 and SDNN behave similarly, authors should present only RMSSD and SDNN. This change must be carried out throughout the manuscript. In non-linear analysis, you can add SampEn if you have the data.
How do you explain that there is no difference between ultra-short measurements in SDNN and RMSSD in table 2 but you observe them in figure 2 in Log SDNN and Log RMSSD? The same is for frequency analysis in table 2 and figure 3.
Figure 4 makes no sense and should be removed.
The results section is well done but it may be too long. Validity analysis with Bland-Altman is correct but makes the manuscript very long. It does not make in wrong but too long.
Discussion should be edited considering the corrections made in the introduction
Why isn't table 5 in the results section?
Author Response
Reviewer 1
I have read this manuscript with interest. It can be suitable for publication in sensors. However, there are some conceptual issues regarding HRV that authors must correct before publishing their manuscript. Once the corrections are made, the manuscript will present quality enough to be published.
Comment 1:
L48-49. Authors state that LF reflects both sympathetic and Parasympathetic nervous activity. How only one variable could represent two opposite things? In addition, LF has been shown not to reflect any branch of nervous activity, thus this statement is not solid and must be deleted. It happens also in lines 51-52 where the authors state the relationship of HF and LF with baroreflexes. Authors should check this reference.
Martelli, D., Silvani, A., McAllen, R. M., May, C. N., &Ramchandra, R. (2014). The low frequency power of heart rate variability is neither a measure of cardiac sympathetic tone nor of baroreflex sensitivity. American Journal of Physiology-Heart and Circulatory Physiology, 307 (7), H1005-H1012.
Response 1:
We appreciated your helpful comment. We have carefully read the above article. As reviewer 1 pointed out, we agree that LF has been indicated to do not reflect sympathetic nervous activity, and the relationship between the frequency component of HRV and baroreflex have not yet been fully agreed upon. Thus, the corresponding statement have been modified as bellow.
Line 46-51. Generally, the frequency band of 0.04–0.15 Hz is defined as the low-frequency (LF) com-ponent. The LF has been reported to be related to baroreflex [14, 15] but have not yet been fully agreed upon [16]. The 0.15–0.4 Hz frequency band is defined as the high-frequency (HF) component and is believed to reflect parasympathetic nervous activity. The HF components is also well known to reflect the periodicity of respiration [17].
[Reference]
- Moak, J. P.; Goldstein, D. S.; Eldadah, B. A.; Saleem, A.; Holmes, C.; Pechnik, S.; Sharabi, Y., Supine low-frequency power of heart rate variability reflects baroreflex function, not cardiac sympathetic innervation. Heart Rhythm 2007, 4, (12), 1523-9.
- Rahman, F.; Pechnik, S.; Gross, D.; Sewell, L.; Goldstein, D. S., Low frequency power of heart rate variability reflects baroreflex function, not cardiac sympathetic innervation. Clinical Autonomic Research 2011, 21, (3), 133-141.
- Martelli, D.; Silvani, A.; McAllen, R. M.; May, C. N.; Ramchandra, R., The low frequency power of heart rate variability is neither a measure of cardiac sympathetic tone nor of baroreflex sensitivity. Am J Physiol Heart Circ Physiol 2014, 307, (7), H1005-12.
- Penttilä, J.; Helminen, A.; Jartti, T.; Kuusela, T.; Huikuri, H. V.; Tulppo, M. P.; Coffeng, R.; Scheinin, H., Time domain, geometrical and frequency domain analysis of cardiac vagal outflow: effects of various respiratory patterns. Clinical Physiology 2001, 21, (3), 365-376.
Comment 2:
L 57-58. Authors affirm that SD1 and SD2 are approaches for the analysis of non-linear dynamics. This statement is false as SD1and SD2 are linear and within the time-domain. Non-linear analysis can be performed by the use of fractal analysis and entropy (Sample entropy mainly). The explanation of the Poincaré Plot in L59-61 is correct but it is not a non-linear analysis. Authors should check this reference.
Ciccone, A. B., Siedlik, J. A., Wecht, J. M., Deckert, J. A.,Nguyen, N. D., & Weir, J. P. (2017). Reminder: RMSSD and SD1are identical heart rate variability metrics. Muscle & nerve, 56 (4),674-678.
Response 2:
We learned a great deal from the above article. As you suggestion, we have removed Poincaré Plot from this manuscript. Instead, we measured SampEn and added in this manuscript. The corresponding statement have been modified as bellow.
Line 54-55. The representative applied approaches include Detrended fluctuation analysis (DFA) and sample entropy (SampEn).
Comment 3:
Authors must mention the multiscale entropy and time irreversibility analysis as a kind of non-linear method of analysis. You can mention these studies.
Abellan-Aynes, O., Naranjo-Orellana, J., Manonelles, P., & Alacid, F. (2020). MULTISCALE ENTROPY AND MULTISCALETIME IRREVERSIBILITY ANALYSIS OF RR TIME SERIESDEPENDING ON AMBIENT TEMPERATURE. Journal of Mechanics in Medicine and Biology, 20 (05), 2050029.
Costa, M., Goldberger, A. L., & Peng, C. K. (2005). Multiscale entropy analysis of biological signals.
Physical review E, 71 (2),021906.
Response 3:
We have mentioned the multiscale entropy and time irreversibility analyses in introduction section, as below.
Line 57-62. SampEn can quantify the complexity of the time series data and has been demonstrated to be reliable for short data set. Recently, the multiscale entropy with SampEn and the analysis of time irreversibility have been proposed as non-linear analysis, and these have been demonstrated to be able to detect a loss of complexity of RR time series with congestive heart failure, aging and external stimulus [19, 20].
[Reference]
- Costa, M.; Goldberger, A. L.; Peng, C. K., Multiscale entropy analysis of biological signals. Physical Review E 2005, 71, (2).
- Abellan-Aynes, O.; Naranjo-Orellana, J.; Manonelles, P.; Alacid, F., MULTISCALE ENTROPY AND MULTISCALE TIME IRREVERSIBILITY ANALYSIS OF RR TIME SERIES DEPENDING ON AMBIENT TEMPERATURE. Journal of Mechanics in Medicine and Biology 2020, 20, (05), 2050029.
Comment 4:
L85. It is true that exercise reduces HRV drastically. From moderate intensity, SDNN and RMSSD can remain close to zero. You should add some citations to state this, I recommend these studies.
Abellán-Aynés, O., López-Plaza, D., Alacid, F., Naranjo-Orellana, J., & Manonelles, P. (2019). Recovery of heart rate variability after exercise under hot conditions: the effect of relative humidity.
Wilderness & Environmental Medicine, 30 (3), 260-267.
Michael, S., Graham, K. S., & Davis, G. M. (2017). Cardiac autonomic responses during exercise and post-exercise recovery using heart rate variability and systolic time intervals—are view. Frontiers in physiology, 8, 301.
Response 4:
We appreciate your helpful suggestion. We added these studies as citation to the below sentence.
Line 84-86. For example, the values for the time- and frequency-domain parameters of HRV decrease progressively from low to moderate exercise intensity [35, 36]
[Reference]
- Abellán-Aynés, O.; López-Plaza, D.; Alacid, F.; Naranjo-Orellana, J.; Manonelles, P., Recovery of Heart Rate Variability After Exercise Under Hot Conditions: The Effect of Relative Humidity. Wilderness & Environmental Medicine 2019, 30, (3), 260-267.
- Michael, S.; Graham, K. S.; Davis, G. M. O., Cardiac Autonomic Responses during Exercise and Post-exercise Recovery Using Heart Rate Variability and Systolic Time Intervals-A Review. Front Physiol 2017, 8, 301.
Comment 5:
Section "2. Research methodology" must be integrated into the material and methods section.
Response 5:
We integrated “Research methodology" into the material and methods section.
Comment 6:
L183-184. SD1 and SD2 as non-linear. The changes regarding this issue must be correct throughout the whole manuscript.
Response 6:
We have modified corresponding sentence as below, and the regarding sentence have been correct throughout the whole manuscript.
Line 182-186. In addition to the frequency-domain, the time-domain (SDNN and RMSSD) and non-linear parameters (DAF α1, α2 and SampEn) were calculated using Kubios HRV software (version 4.0, Biosignal Analysis and Medical Imaging Group, Department of Physics, University of Kuopio, Kuopio, Finland).
Comment 7:
L189. Why didn't you measure SampEn?
Response 7:
We measured SampEn and add in result section, as bellow.
Line 271-273. SampEn decreased at 50%VO2peak compared to 20%VO2peak but increased at 80%VO2peak compared to 50%VO2peak in the 180-sec, 120-sec, 90-sec, 60-sec segments (p < 0.05).
Figure 4. Changes of each ultra-short-term segment of non-linear HRV parameters during exercise. DFA, detrended fluctuation analysis; SampEn, sample entropy; log, logarithmic; sec, seconds; VO2peak, peak oxygen uptake. The middle line indicates the mean value. The top and bottom lines indicate the SD. The plot indicates personal values. Note that there is missing data (see the Materials and Methods section). *, p < 0.05 vs. 20%VO2peak; †, p < 0.05 vs. 50%VO2peak.
Line 323-325. SampEn exhibited a high correlation between the 180-sec and 120-sec segment at 20% and 50%VO2peak (≧0.8; p < 0.05).
Line 351-353. The SampEn between ultra-short-term segments were small bias (effect size; ≦0.5) except for the 30-sec segment at 50%VO2peak.
Comment 8:
Table 2. As SD1 and RMSSD are exactly the same variable (RMMSD=SD1/Root square (2)) and SD2 and SDNN have similarly, authors should present only RMSSD and SDNN. This change must be carried out throughout the manuscript. In non-linear analysis, you can add SampEn if you have the data.
Response 8:
Please see Comment 2, 3 and 7 of Reviewer 1 regarding this response.
Comment 9:
How do you explain that there is no difference between ultra-short measurements in SDNN and RMSSD in table 2 but you observe them in figure 2 in Log SDNN and Log RMSSD? The same is for frequency analysis in table 2 and figure 3.
Response 9:
We appreciate your comment. The absolute values of most HRV were distributed non-normally excluding the non-linear parameters. Thus, we have conducted statistical analysis using the log-transformed HRV values but did not for the absolute values. However, we believe it is important to present the absolute values of HRV parameters before log-transformed, the mean RR and HR. Accordingly, in the manuscript before revision, the aim have been to present the absolute values of HRV parameters in Table 2 as reference values. However, as reviewer 1 pointed out, Table 2 can be misleading with respect to interpretation of results. Therefore, the absolute values of HRV parameters before log-transformed, the mean RR and HR have moved to supplemental material (Tabel S1). In addition, we modified specific sentences in the result section of manuscript as below:
Line 259-260. Figures 2, 3, and 4 show mean HRV values (log) during exercise. Refer to Table S1 for the absolute values of HRV parameters, the mean RR and HR.
Comment 10:
Figure 4 makes no sense and should be removed.
Response 10:
As mentioned above comment 9, Supplemental material (Tabel S1) values were not conducted statistical analysis. Thus, Figure 4 has not been removed to show the results of statistical analysis in non-linear parameters.
Comment 11:
The results section is well done but it may be too long. Validity analysis with Bland-Altman is correct but makes the manuscript very long. It does not make in wrong but too long.
Response 11:
We appreciate your helpful comment. We included the Bland-Altman plot in supplemental material (Figure S1–5). In addition, we modified specific sentences in result section of the manuscript as below:
Line 335-337. The Bland-Altman plots for the 180-sec segment and other ultra-short-term HRV segments (120 sec, 90 sec, 60 sec, and 30 sec) are indicated in Figures S1–5.
Comment 12:
Discussion should be edited considering the corrections made in the introduction.
Response 12:
We removed the sentences regarding to the Poincaré Plot in the discussion section.
Comment 13:
Why isn't table 5 in the results section?
Response 13:
Table 4 (the pre-revision was Table 5) have included in the result section as bellow.
Line 327-328. Based on these results, the acceptable minimum time segment of HRV analysis according to the correlation coefficient were summarized in Table 4.
Line 354-355. Based on these results, the acceptable minimum time segment of HRV analysis according to the effect size were summarized in Table 4.
Reviewer 2 Report (New Reviewer)
Overall quality of current paper is extremely good but before considering acceptance by editors I think MINOR REVISIONS are necessary.
Among POINTs of WEAKNESSES we can list:
English language to be checked; sentences such as at lines 16-17 and/or 33-35 should be totally re-written.
Quality of tables (especially number 2) might be ameliorated.
Finally, I'd like to suggest the following articles in order to insert them in references and/or contributing to general improvement of discussion:
--Sensors (Basel). 2022 Nov 25;22(23):9149. doi: 10.3390/s22239149
--J Strength Cond Res. 2020 Oct;34(10):2974-2981. doi: 10.1519/JSC.0000000000002910
Best regards.
Author Response
Title: The Validity of Ultra-Short-Term Heart Rate Variability during Cycling Exercise
Journal: Sensors
We thank the reviewers for carefully reading our manuscript and providing useful comments. We have substantially revised our manuscript in accordance with the reviewers’ comments.
In addition, one table and five figures have moved to supplementary material.
Our responses to the reviewers’ comments are given below.
-------------------------------------------------------------------------------------------------------
Reviewer 2
Overall quality of current paper is extremely good but before considering acceptance by editors I think MINOR REVISIONS are necessary.
Comment 1:
English language to be checked; sentences such as at lines 16-17 and/or 33-35 should be totally re-written.
Response 1:
Thank you for your helpful comment. the proofreading has been carried out by native before 1st submit. We attached the certificate of editing as below.
However, we modified specific sentences as below:
Line 16-17. Ultra-short-term heart rate variability (HRV) has been validated in the resting state, but its validity during exercise is unclear.
Line 32-34. Heart rate variability (HRV) is the fluctuation of the instantaneous heartbeat over time [1]. HRV is traditionally used for non-invasive assessments of cardiac autonomic nervous activity [2].
Comment 2:
Quality of tables (especially number 2) might be ameliorated.
Response 2:
Please see comment 9 of Reviewer 1.
Comment 3:
Finally, I'd like to suggest the following articles in order to insert them in references and/or contributing to general improvement of discussion:
--Sensors (Basel). 2022 Nov 25;22(23):9149. doi:10.3390/s22239149
--J Strength Cond Res. 2020 Oct;34(10):2974-2981. doi:10.1519/JSC.0000000000002910
Response 3:
We thank for you helpful suggestion. We added the above articles in discussion section and modified the corresponding sentence as bellow.
Line 391-392. The previous study has demonstrated similar results in stress test (a head-up tilt) [53].
[Reference]
- Volpes, G.; Barà, C.; Busacca, A.; Stivala, S.; Javorka, M.; Faes, L.; Pernice, R., Feasibility of Ultra-Short-Term Analysis of Heart Rate and Systolic Arterial Pressure Variability at Rest and during Stress via Time-Domain and Entropy-Based Measures. Sensors 2022, 22, (23), 9149.
Line 431-434. The previous study for athletes has been suggested to measure the ultra-short-term HRV before training in multiple days for monitoring condition and training adaptation [58]. Accordingly, the realistic measurement time seems to be an important factor in spreading HRV assessment.
[Reference]
- Nakamura, F. Y.; Antunes, P.; Nunes, C.; Costa, J. A.; Esco, M. R.; Travassos, B., Heart Rate Variability Changes From Traditional vs. Ultra-Short-Term Recordings in Relation to Preseason Training Load and Performance in Futsal Players. J Strength Cond Res 2020, 34, (10), 2974-2981.
Reviewer 3 Report (New Reviewer)
Authors investigated the possibility of ultra-short-term heart rate variability measurements during exercise. The study is interesting and in line with journal's aims and scope.
My main concern are the conclusions derived from the results. Authors suggest that they can determine the valid time duration for HRV analysis across all the exercise intensities. This finding would contribute to accurate HRV analysis in non-static conditions, such as exercise. However, the HRV depends on the type of activity. The duration of recording for valid HRV determination depends on the state (rest or active) and if the subject is in the active state depends on the type and intensity of the activity.
Therefore, in my opinion, based on the current results the authors can claim that the ultra-short-term heart rate variability is feasible in the concrete task, and that they determined some optimal time intervals for across exercise intensities during the concrete task. However, any generalization of these results is too far-fetched and should be removed from the conclusions.
Author Response
Title: The Validity of Ultra-Short-Term Heart Rate Variability during Cycling Exercise
Journal: Sensors
We thank the reviewers for carefully reading our manuscript and providing useful comments. We have substantially revised our manuscript in accordance with the reviewers’ comments.
In addition, one table and five figures have moved to supplementary material.
Our responses to the reviewers’ comments are given below.
-------------------------------------------------------------------------------------------------------
Reviewer 3
Authors investigated the possibility of ultra-short-term heart rate variability measurements during exercise. The study is interesting and in line with journal's aims and scope.
My main concern are the conclusions derived from the results.
Authors suggest that they can determine the valid time duration for HRV analysis across all the exercise intensities. This finding would contribute to accurate HRV analysis in non-static conditions, such as exercise. However, the HRV depends on the type of activity. The duration of recording for valid HRVdetermination depends on the state (rest or active) and if the subject is in the active state depends on the type and intensity of the activity.
Therefore, in my opinion, based on the current results the authors can claim that the ultra-short-term heart rate variability is feasible in the concrete task, and that they determined some optimal time intervals for across exercise intensities during the concrete task. However, any generalization of these results is too far-fetched and should be removed from the conclusions.
Response:
We appreciate for carefully reading our manuscript, and for your great suggestion. In this study, although we examined HRV in three exercise intensities, the type of exercise has been limited in cycling. Therefore, we agree your opinion. Accordingly, we have modified the conclusions as bellow.
Line 463-465. The validity of ultra-short-term HRV was different for exercise intensities. However, the ultra-short-term HRV is feasible in the cycling exercise and determined some optimal time duration for HRV analysis for across exercise intensities during cycling exercise.
Round 2
Reviewer 1 Report (New Reviewer)
In the previous review, I asked for substantial changes to the authors. The authors have carried out a good work in the edition of the paper. Many important concerns have been addressed and the paper is now suitable for publication in this journal.
From my point of view, this article does not need any further changes.
Congratulations
Author Response
Thank you very much for your excellent comments for our manuscript.
Reviewer 3 Report (New Reviewer)
There is still an open issue regarding my previous comment that the authors adressed.
Authors claim that "ultra-short-term HRV is feasible in the cycling exercise and determined some optimal time duration for HRV analysis for across exercise intensities during cycling exercise." However, they do not clearly justify their approach. Since they adapted and restricted their method to cycling exercise, they should very clearly state the added value of their method for the cycling exercise, i.e. what are the applications of their method during cycling in particular.
Author Response
Title: The Validity of Ultra-Short-Term Heart Rate Variability during Cycling Exercise
Journal: Sensors
We thank the reviewers for carefully reading our manuscript and providing useful comments. We have substantially revised our manuscript in accordance with the reviewers’ comments.
Our responses to the reviewers’ comments are given below.
------------------------------------------------------------------------------
Reviewer 3
Authors claim that "ultra-short-term HRV is feasible in the cycling exercise and determined some optimal time duration for HRV analysis for across exercise intensities during cycling exercise." However, they do not clearly justify their approach. Since they adapted and restricted their method to cycling exercise, they should very clearly state the added value of their method for the cycling exercise, i.e. what are the applications of their method during cycling in particular.
Response: We appreciate for carefully reading our manuscript. We have revised the conclusion section and the abstract corresponding to the conclusion as follows.
Line 26-28. However, the ultra-short-term HRV is feasible in the cycling exercise, and we determined some optimal time duration for HRV analysis for across exercise intensities during the incremental cycling exercise.
Line 464-467. However, the ultra-short-term HRV is feasible in the cycling exercise, and we determined some optimal time duration for HRV analysis for across exercise intensities during the incremental cycling exercise in particular.
This manuscript is a resubmission of an earlier submission. The following is a list of the peer review reports and author responses from that submission.
Round 1
Reviewer 1 Report
The paper entitled "The validity of the ultra-short term in heart rate variability during cycling exercise" is well written. Presented research results are up to the academic standards. The paper has a lot of merit and should be recommended for publication, but after correcting some important issues from the point of view of the quality of "Sensors" journal.
1. I would like to suggest authors to extension of the introduction so that this section can fully present the current state of art.
2. I also propose to add nomenclature (description of used symbols and abbreviations) to improve the readability of the article.
3. Figure 2 presented in the paper is not readable.
4. Although the topic and research results presented in the paper is current and interesting, the references section in the peer-revied paper is poor. The references section does not contain the current state of knowledge in the scope presented in the peer-revied paper.
5. Please specify the section on research methodology.
6. Equations should always be written on a separate line, and never placed in the text (such as line 188).
7. It seems that the research sample for which statistical research has been carried out with the use of ANOVA is relatively small. Thus, there remains the question of the reliability of the conclusions. Please comment.
Author Response
Title: The validity of the ultra-short term in heart rate variability during cycling exercise
Journal: Sensors
We thank the reviewers for carefully reading our manuscript and providing useful comments. We have substantially revised our manuscript in accordance with the reviewers’ comments.
Our responses to the reviewers’ comments are given below.
-------------------------------------------------------------------------------------------------------------------------
REVIEWER 1
The paper entitled "The validity of the ultra-short term in heart rate variability during cycling exercise" is well written. Presented research results are up to the academic standards. The paper has a lot of merit and should be recommended for publication, but after correcting some important issues from the point of view of the quality of "Sensors" journal.
Comment 1. I would like to suggest authors to extension of the introduction so that this section can fully present the current state of art.
Response 1. We thank the reviewer for this pertinent suggestion. We contained the current state of art about the ultra-short term HRV in introduction section as follow:
Line 63: Recently, although there is little agreement as to the acceptable time periods of recording and analysis in the ultra-short-term HRV, there are many studies that examined the validity and reproducibility of ultra-short-term HRV [18-24]. Burma et al. concluded that 300 seconds (sec) were acceptable to obtain accurate the relative frequency-domain HRV parameters (the combination of very low frequency [0.0033-0.04 Hz], LF and HF summated to 100) in a standing position [20]. In the resent review, minimum time period of the LF and HF absolute power to estimate 5-min HRV were shown to be 60 sec and 180 sec, respectively [21].
[References]
- Thong, T.; Li, K.; McNames, J.; Aboy, M.; Goldstein, B. In Accuracy of ultra-short heart rate variability measures, Proceedings of the 25th Annual International Conference of the IEEE Engineering in Medicine and Biology Society (IEEE Cat. No. 03CH37439), 2003; IEEE: 2003; pp 2424-2427.
- Schroeder, E. B.; Whitsel, E. A.; Evans, G. W.; Prineas, R. J.; Chambless, L. E.; Heiss, G., Repeatability of heart rate variability measures. J Electrocardiol 2004, 37, (3), 163-72.
- Burma, J. S.; Graver, S.; Miutz, L. N.; Macaulay, A.; Copeland, P. V.; Smirl, J. D., The validity and reliability of ultra-short-term heart rate variability parameters and the influence of physiological covariates. J Appl Physiol (1985) 2021, 130, (6), 1848-1867.
- Shaffer, F.; Meehan, Z. M.; Zerr, C. L., A Critical Review of Ultra-Short-Term Heart Rate Variability Norms Research. Front Neurosci 2020, 14, 594880.
- Nussinovitch, U.; Elishkevitz, K. P.; Katz, K.; Nussinovitch, M.; Segev, S.; Volovitz, B.; Nussinovitch, N., Reliability of Ultra-Short ECG Indices for Heart Rate Variability. Annals of Noninvasive Electrocardiology 2011, 16, (2), 117-122.
- Hoffmann, B.; Flatt, A. A.; Silva, L. E. V.; Mlynczak, M.; Baranowski, R.; Dziedzic, E.; Werner, B.; Gasior, J. S., A Pilot Study of the Reliability and Agreement of Heart Rate, Respiratory Rate and Short-Term Heart Rate Variability in Elite Modern Pentathlon Athletes. Diagnostics (Basel) 2020, 10, (10).
- Baek, H. J.; Cho, C. H.; Cho, J.; Woo, J. M., Reliability of ultra-short-term analysis as a surrogate of standard 5-min analysis of heart rate variability. Telemed J E Health 2015, 21, (5), 404-14.
Comment 2. I also propose to add nomenclature (description of used symbols and abbreviations) to improve the readability of the article.
Response 2. Thank you for your helpful comment. We appropriately added the corresponding description of used symbols and abbreviations in the text, figures legends and tables.
Comment 3. Figure 2 presented in the paper is not readable.
Response 3. Thank you for your point out. We revised Figure 2 and figure legend as follow:
Figure 2. Changes of each ultra-short-term segment HRV parameters during exercise. HF, high-frequency power; LF, low-frequency power; log, logarithmic; sec, seconds; VO2peak, peak oxygen uptake. The middle line indicates the mean value. The top and bottom lines indicate the SD. The plot indicates personal values. Note that there is missing data at 20%VO2peak (see the Materials and Methods section). a, p < 0.05 vs. 20%VO2peak; b, p < 0.05 vs. 50%VO2peak.
Comment 4. Although the topic and research results presented in the paper is current and interesting, the references section in the peer-revied paper is poor. The references section does not contain the current state of knowledge in the scope presented in the peer-revied paper.
Response 4. We added the current peer-revied paper about the ultra-short term HRV in the references section. The added articles are as bellow:
- Hoffmann, B.; Flatt, A. A.; Silva, L. E. V.; Mlynczak, M.; Baranowski, R.; Dziedzic, E.; Werner, B.; Gasior, J. S., A Pilot Study of the Reliability and Agreement of Heart Rate, Respiratory Rate and Short-Term Heart Rate Variability in Elite Modern Pentathlon Athletes. Diagnostics (Basel) 2020, 10, (10).
- Baek, H. J.; Cho, C. H.; Cho, J.; Woo, J. M., Reliability of ultra-short-term analysis as a surrogate of standard 5-min analysis of heart rate variability. Telemed J E Health 2015, 21, (5), 404-14.
- Rossi, A.; Pedreschi, D.; Clifton, D. A.; Morelli, D., Error Estimation of Ultra-Short Heart Rate Variability Parameters: Effect of Missing Data Caused by Motion Artifacts. Sensors (Basel) 2020, 20, (24).
- Wu, L.; Shi, P.; Yu, H.; Liu, Y., An optimization study of the ultra-short period for HRV analysis at rest and post-exercise. J Electrocardiol 2020, 63, 57-63.
- Forner-Llacer, F.; Aranda-Malaves, R.; Aranda Malavés, R.; Calleja-Gonzalez, J.; Perez-Turpin, J. A.; Gonzalez-Rodenas, J., Minimal Stabilization Time for Ultra-short Heart Rate Variability Measurements in Professional Soccer. International Journal of Sports Medicine 2020, 41, (14), 1032-1038.
- Krejčí, J.; Botek, M.; McKune, A. J., Stabilization period before capturing an ultra-short vagal index can be shortened to 60 s in endurance athletes and to 90 s in university students. PLOS ONE 2018, 13, (10), e0205115.
- Pereira, L. A.; Flatt, A. A.; Ramirez-Campillo, R.; Loturco, I.; Nakamura, F. Y., Assessing Shortened Field-Based Heart-Rate-Variability-Data Acquisition in Team-Sport Athletes. International Journal of Sports Physiology and Performance 2016, 11, (2), 154-158.
- Esco, M. R.; Flatt, A. A., Ultra-short-term heart rate variability indexes at rest and post-exercise in athletes: evaluating the agreement with accepted recommendations. J Sports Sci Med 2014, 13, (3), 535-41.
- Chen, Y. S.; Pagaduan, J. C.; Bezerra, P.; Crowley-McHattan, Z. J.; Kuo, C. D.; Clemente, F. M., Agreement of Ultra-Short-Term Heart Rate Variability Recordings During Overseas Training Camps in Under-20 National Futsal Players. Front Psychol 2021, 12, 621399.
- Bourdillon, N.; Schmitt, L.; Yazdani, S.; Vesin, J. M.; Millet, G. P., Minimal Window Duration for Accurate HRV Recording in Athletes. Front Neurosci 2017, 11, 456.
Comment 5. Please specify the section on research methodology.
Response 5. We appreciate the reviewer’s great suggestion, and it is very important point. We add the section about research methodology in the present study as follow:
Line 88: 2. Research methodology
The goal of the present study is to suggest the valid time segment in HRV analysis, particularly the frequency domain analysis, in exercise conditions. In the present study, we examined using the R–R intervals that collected during an incremental exercise test. The novelty is to examine the validity of the five ultra-short-term segments (30 sec, 60 sec, 90 sec, 120 sec, 180 sec segments) on each of three exercise intensity (low, moderate, and high intensity). The present study examined the validity according to the following step. First, in each five segments, we confirmed the changes of mean values of frequency do-main of HRV with increasing exercise intensity, while comparing with physiological parameters changes. Secondly, at each exercise intensity, 180 seconds was used as a reference to compare with other time segments to confirm the correlation and determination coefficient. Finally, we revealed the difference and bias between 180 sec and other time segments at each exercise intensity. This finding would contribute HRV analysis during in non-static conditions, such as during exercise.
Comment 6. Equations should always be written on a separate line, and never placed in the text (such as line 188).
Response 6. We have written the equation on a separate line (Line 217) from the text.
Comment 7. It seems that the research sample for which statistical research has been carried out with the use of ANOVA is relatively small. Thus, there remains the question of the reliability of the conclusions. Please comment.
Response 7. We appreciate for carefully reading our manuscript. We confirmed the sample size using the partial η2 that calculated by ANOVA analysis. The smallest partial η2 in all ANOVA analysis in the present study was 0.216, and the effect size was 0.525. We calculated the total sample size when we set the parameters as follow: the effect size = 0.525, αerror prob = 0.05, power (1-β error prob) = 0.8, Corr among rep measures = 0.5, and Nonsphericition correction ε = 0.5. As the result, the total sample size was 12. Therefore, we considered that the research sample in present study is sufficient.
Reviewer 2 Report
This study aimed to investigate the effects of cycling exercise on heart rate variability. I have the following concerns.
- What is the novelty of this study although several studies to investigate the effect of exercise on HRV have been reported earlier? Please add a paragraph about the contribution of this article and the structure of the study at the end part of the Introduction section.
- ECG is highly sensitive to the powerline, muscular artifacts. In ECG data preprocessing, authors need to mention how you handle AC power, movement, and EMG artifacts in ECG signals.
- Authors should review the HRV studies, such as mental workload, stress, emotion, brain-stimulation. HRV are investigated for brain-stimulation in article, quantifying physiological biomarkers of a microwave brain stimulation device, and in article, big-ecg: cardiographic predictive cyber-physical system for stroke management.
- Authors should add a figure showing the experimental protocol.
- Authors should mention the HRV analysis methodology in details, such as epoch, EMG artifact removal, R-peak detection algorithms. Authors should add references of case studies mentioned above.
- introduction, authors described HF power refers regulatory function of the vagus nerve; and LF power, partial contributions from both pathetic and parasympathetic nervous system activities. Several recent studies don’t fully agree with this hypothesis. As example,
G. E. Billman, "The LF/HF ratio does not accurately measure cardiac sympatho-vagal balance," Frontiers in physiology, vol. 4, p. 26, 2013.
- I recommend using non-linear HRV analysis.
- Authors need to discuss and comments about this controversy in discussion section.
9. Authors should add a table listing all results for better readability.
10. Authors should discuss the strength and weaknesses of the proposed method with other recent studies adding a comparative table in the discussion section.
- From the writing point of view, the manuscript must be checked for typos and the grammatical issues should be improved.
Author Response
Title: The validity of the ultra-short term in heart rate variability during cycling exercise
Journal: Sensors
We thank the reviewers for carefully reading our manuscript and providing useful comments. We have substantially revised our manuscript in accordance with the reviewers’ comments.
Our responses to the reviewers’ comments are given below.
-------------------------------------------------------------------------------------------------------------------------
REVIEWER 2
This study aimed to investigate the effects of cycling exercise on heart rate variability. I have the following concerns.
Comment 1. What is the novelty of this study although several studies to investigate the effect of exercise on HRV have been reported earlier? Please add a paragraph about the contribution of this article and the structure of the study at the end part of the Introduction section.
Response 1. We appreciate for carefully reading our manuscript, and for your great suggestion. We agree with your comment that the effect of exercise on HRV have been reported in many previous studies. However, the aim of this study was to reveal the consentaneity, the difference and bias of frequency components of HRV among the different time segment (30–180 sec) of HRV analysis, particularly, the novelty was to examine each three exercise intensities (Low, moderate, and high intensity). Thus, we have considered that this finding can contribute to HRV analysis during in non-static condition such as the physical activity and exercise.
We added the section of research methodology and include the novelty, structure, and contribution of this article in that section (Line 88). (See comment 5 for reviewer 1)
Comment 2. ECG is highly sensitive to the powerline, muscular artifacts. In ECG data preprocessing, authors need to mention how you handle AC power, movement, and EMG artifacts in ECG signals.
Response 2. Thank you for your point out. We agree the handling of these factors is important. However, we have used the existing products “Reflex-Meijin” software program (Crosswell, Yokohama, Japan) which automatically and immediately calculated R–R intervals from ECG. We checked that of site, but the information of how to handle was not disclosed. Instead, we added the information about the procedures of excluding the artifacts and abnormal beats using “Memcalc/Win” software program (GMS, Tokyo, Japan) before the spectral analysis as follow:
Line 159: Before the frequency analysis, the artifacts and abnormal beats were excluded using “Memcalc/Win” software program (GMS, Tokyo, Japan) according to the following criteria: 1) HR was below 30 bpm and above 220 bpm, 2) outlier which was detected by comparing adjacent R–R data, 3) the validity was not certified among the successive three R–R data.
Comment 3. Authors should review the HRV studies, such as mental workload, stress, emotion, brain-stimulation. HRV are investigated for brain-stimulation in article, quantifying physiological biomarkers of a microwave brain stimulation device, and in article, big-ecg: cardiographic predictive cyber-physical system for stroke management.
Response 3. Thank you for the suggestion. While we agree that HRV could become physiological biomarkers and HRV is affected by the mental workload, stress, and emotion, we considered that it moves the main point away from the aim of present study. Therefore, we have not mention the HRV studies, such as mental workload, stress, emotion, brain-stimulation in the revised manuscript.
Comment 4. Authors should add a figure showing the experimental protocol.
Response 4. Thank you for your helpful suggestion. We add the experimental protocol in Figure 1.
Figure 1. The experimental protocol and the adoption of HRVs corresponding to each exercise intensity. The black range in time segment of HRV analysis indicates the time length of HRV analysis from each exercise intensity (20%, 50%, or 80% VO2peak). All HRV analysis within the individual participant were performed using the same R–R intervals recording data. Up arrow is the sampling point of blood. Ex, exercise; HRV, heart rate variability; sec, seconds; VO2peak, peak oxygen uptake.
Comment 5. Authors should mention the HRV analysis methodology in details, such as epoch, EMG artifact removal, R-peak detection algorithms. Authors should add references of case studies mentioned above.
Response 5. Thank you for your helpful comment. We have mentioned the additional information of HRV analysis. Please see comment 2 for Reviewer 2.
Comment 6. Introduction, authors described HF power refers regulatory function of the vagus nerve; and LF power, partial contributions from both pathetic and parasympathetic nervous system activities. Several recent studies don’t fully agree with this hypothesis. As example,
- E. Billman, "The LF/HF ratio does not accurately measure cardiac sympatho-vagal balance," Frontiers in physiology, vol. 4, p. 26, 2013.
Response 6. We agree with your comment. We added that the concern about the hypothesis of HF and LF in the introduction section (Line 43) as below:
Line 43: Generally, the frequency band of 0.04–0.15 Hz is defined as the low-frequency (LF) com-ponent, and it reflects both sympathetic and parasympathetic nervous activity. The 0.15–0.4 Hz frequency band is defined as the high-frequency (HF) component and is believed to reflect the parasympathetic nervous activity. However, those hypothesis about the relationship between LF, HF and autonomic nervous system have not fully agreed currently [14].
[Reference]
- Billman, G. E., The LF/HF ratio does not accurately measure cardiac sympatho-vagal balance. Front Physiol 2013, 4, 26.
Comment 7. I recommend using non-linear HRV analysis.
Response 7. We agree that the non-linear HRV analysis is one of the important aspects in HRV analysis, but in this article, we prefer to focus on the frequency analysis of HRV, and so we will leave the non-linear HRV analysis to other researches.
Comment 8. Authors need to discuss and comments about this controversy in discussion section.
Response 8. Thank you for your helpful suggestion. We have discussed about the controversy of ultra-short-term HRV in discussion section as follow:
Line 382: There are many previous studies examined the validity of ultra-short-term HRV in the resting condition, but there is little agreement about the required length of HRV analysis. In the frequency domain analysis, one study reported that the acceptable shorter segment was 20 sec for HF component [24]. On the other hands, there was the previous study that the 300 sec segment was acceptable [20]. These disagreement of the ultra-short-term HRV could be due to the artifacts, physiological influences, measurement postures, and so on [20, 37]. While the controversy of the ultra-short-term HRV in the resting condition is remained, the examination under the non-static condition such as exercise is also needed, because the practical uses HRV assessment is extend. The resent one study concluded that the minimum time needed for the ultra-short-term HRV analysis was 10–180 sec in exercise condition [25]. However, the exercise intensity was not considerate in this study, de-spite HRV non-linearly change with exercise intensity increased [10, 26]. On the other hands, the present study was examined that the exercise condition was separated three stages (low, moderate, and high intensity). This is strength point of the present study, and it is the first article that presented the validity of ultra-short-term HRV considering the different exercise intensity. However, the presented the validity of ultra-short-term HRV in this report is limited the frequency domain of HRV analysis. In many previous studies, the validity of ultra-short-term HRV in the time domain and the non-linear analysis have been also examined [21, 38-44]. The acceptable minimum time for ultra-short-term HRV analysis has been shown to be different among the frequency domain, the time domain, and the non-linear HRV analysis [20-22, 24, 25, 45]. The further study including with the time domain and non-linear HRV analysis in exercise condition are also needed because these two analyses are also major methods of HRV.
[Reference]
- Burma, J. S.; Graver, S.; Miutz, L. N.; Macaulay, A.; Copeland, P. V.; Smirl, J. D., The validity and reliability of ultra-short-term heart rate variability parameters and the influence of physiological covariates. J Appl Physiol (1985) 2021, 130, (6), 1848-1867.
- Shaffer, F.; Meehan, Z. M.; Zerr, C. L., A Critical Review of Ultra-Short-Term Heart Rate Variability Norms Research. Front Neurosci 2020, 14, 594880.
- Baek, H. J.; Cho, C. H.; Cho, J.; Woo, J. M., Reliability of ultra-short-term analysis as a surrogate of standard 5-min analysis of heart rate variability. Telemed J E Health 2015, 21, (5), 404-14.
- Kim, J. W.; Seok, H. S.; Shin, H., Is Ultra-Short-Term Heart Rate Variability Valid in Non-static Conditions? Front Physiol 2021, 12, 596060.
- Tanoue, Y.; Komatsu, T.; Nakashima, S.; Matsuda, T.; Michishita, R.; Higaki, Y.; Uehara, Y., The ratio of heart rate to heart rate variability reflects sympathetic activity during incremental cycling exercise. Eur J Sport Sci 2021, 1-10.
- Rossi, A.; Pedreschi, D.; Clifton, D. A.; Morelli, D., Error Estimation of Ultra-Short Heart Rate Variability Parameters: Effect of Missing Data Caused by Motion Artifacts. Sensors (Basel) 2020, 20, (24).
- Wu, L.; Shi, P.; Yu, H.; Liu, Y., An optimization study of the ultra-short period for HRV analysis at rest and post-exercise. J Electrocardiol 2020, 63, 57-63.
- Forner-Llacer, F.; Aranda-Malaves, R.; Aranda Malavés, R.; Calleja-Gonzalez, J.; Perez-Turpin, J. A.; Gonzalez-Rodenas, J., Minimal Stabilization Time for Ultra-short Heart Rate Variability Measurements in Professional Soccer. International Journal of Sports Medicine 2020, 41, (14), 1032-1038.
- Krejčí, J.; Botek, M.; McKune, A. J., Stabilization period before capturing an ultra-short vagal index can be shortened to 60 s in endurance athletes and to 90 s in university students. PLOS ONE 2018, 13, (10), e0205115.
- Pereira, L. A.; Flatt, A. A.; Ramirez-Campillo, R.; Loturco, I.; Nakamura, F. Y., Assessing Shortened Field-Based Heart-Rate-Variability-Data Acquisition in Team-Sport Athletes. International Journal of Sports Physiology and Performance 2016, 11, (2), 154-158.
- Munoz, M. L.; van Roon, A.; Riese, H.; Thio, C.; Oostenbroek, E.; Westrik, I.; de Geus, E. J.; Gansevoort, R.; Lefrandt, J.; Nolte, I. M.; Snieder, H., Validity of (Ultra-)Short Recordings for Heart Rate Variability Measurements. PLoS One 2015, 10, (9), e0138921.
- Esco, M. R.; Flatt, A. A., Ultra-short-term heart rate variability indexes at rest and post-exercise in athletes: evaluating the agreement with accepted recommendations. J Sports Sci Med 2014, 13, (3), 535-41.
- Chen, Y. S.; Pagaduan, J. C.; Bezerra, P.; Crowley-McHattan, Z. J.; Kuo, C. D.; Clemente, F. M., Agreement of Ultra-Short-Term Heart Rate Variability Recordings During Overseas Training Camps in Under-20 National Futsal Players. Front Psychol 2021, 12, 621399.
- Bourdillon, N.; Schmitt, L.; Yazdani, S.; Vesin, J. M.; Millet, G. P., Minimal Window Duration for Accurate HRV Recording in Athletes. Front Neurosci 2017, 11, 456.
Comment 9. Authors should add a table listing all results for better readability.
Response 9. We thank the reviewer for this pertinent suggestion. We add a table listing all results in discussion section.
Comment 10. Authors should discuss the strength and weaknesses of the proposed method with other recent studies adding a comparative table in the discussion section.
Response 10. We thank the reviewer for this pertinent suggestion. We included the above point in discussion section. Please see comment 8 for Reviewer 2.
Comment 11. From the writing point of view, the manuscript must be checked for typos and the grammatical issues should be improved.
Response 11. We appreciate for carefully reading our manuscript. The manuscript has carefully checked and revised.
Round 2
Reviewer 1 Report
The authors took into account the comments I indicated exhaustively. I think that the article can be published in its current form.
Author Response
We thank the reviewer for carefully reading our manuscript and providing useful comments.
Reviewer 2 Report
Several review comments were skipped.
This study has already been reported by several researchers.
Authors didn't report non-linear HRV analysis as suggested.
Extensive comparative discussion with other studies is reported in the discussion section.
Round 3
Reviewer 2 Report
I have no more comments to improve the novelty further.